# Learning to Jointly Understand Visual and Tactile Signals

**Yichen Li**[1], **Yilun Du**[1], **Chao Liu**[1], **Chao Liu**[2], **Francis Williams**[2], **Michael Foshey**[1],
**Ben Eckart**[2], **Jan Kautz**[2], **Joshua B. Tenenbaum**[1], **Antonio Torralba**[1], **Wojciech Matusik**[1]
[1]MIT CSAIL [2]NVIDIA

## Abstract

Modeling and analyzing objects and shapes has been well-studied in the past. However, manipulation of these complex tools and articulated objects remains difficult for autonomous agents. Our human hands, however, are dexterous and adaptive. We can easily adapt a manipulation skill on one object to all objects in the class and to other similar classes. Our intuition comes from that there is a close connection between manipulations and topology and articulation of objects. The possible articulation of objects indicates the types of manipulation necessary to operate the object. In this work, we aim to take a manipulation perspective to understand everyday objects and tools. We collect a multi-modal visual-tactile dataset that contains paired full-hand force pressure maps and manipulation videos. We also propose a novel method to learn a cross-modal latent manifold that allows for cross-modal prediction and discovery of latent structure in different data modalities. We conduct extensive experiments to demonstrate the effectiveness of our method.[‡]

## 1 Introduction

Manipulating complex tools and articulated objects is a crucial yet difficult task for autonomous agents. Existing efforts on modeling and analyzing objects and shapes focus on appearance (Li et al., 2015; Schulz et al., 2017; Li et al., 2020b), topology (Luo et al., 2008; Duan et al., 2008; Hui et al., 2022), and articulation (Li et al., 2020a; Mo et al., 2021; Noguchi et al., 2021; Yi et al., 2018). To motivate a dexterous robotic hand, we study human hand interaction with complex everyday objects that comes in different shapes and forms. A key insight into the robustness and universality of human manipulation is the underlying hand synergy subspace (Santello et al., 2013). We hypothesize that hand force synergy and the articulation of tools are closely connected. In this paper, we aim to construct a computational model that may discover the cross-modal structure shared through the two perceptual modalities, tactile that capture the force pressure of the hand, and vision, which captures the appearance, topology, and articulation of objects.

We focus on tactile and vision, and study ways to discover the underlying structure of force fields on our hands and to connect them with our visual senses. To achieve this, we focus on cross-modal prediction, the ability to synthesize temporal force signals from vision inputs, and predicting plausible vision signals from tactile force sequences. The main challenge lies in the disparity in spatial scale between the signals of vision and tactile and our ability to discover the shared structure across each modality. Our visual perception mechanism digests a complete scene at once, whereas our tactile force senses can only perceive a minute portion of an object at the scale of our hand. This gap in scale imposes a challenge in the mapping mechanism to be a many-to-many mapping between tactile and vision, where similar haptic force senses are plausible for different objects and one object can be manipulated using various forces.

To this end, we collect a cross-modal force-vision dataset. We collect the active full-hand force pressure sequence data using specialized tactile sensor gloves (Sundaram et al., 2019b). We also use a webcam to record visual data of both hand-object manipulation sequences. In total, we collected 2,000,000 frames of paired hand force pressure and video frames over 89 real object instances of different object categories.

In this work, we represent different modalities of information through manifold learning. However, directly using the shared latent space to represent both vision and force will cause the richer data

---

[‡]For further references: https://sites.google.com/view/iclr-submission-force-vision/home?authuser=3

modality to dominate the more local and sparser data modality. To address the difference in scale, we introduce sub-manifold projections to allow different sensory modalities to be represented in their local scale while preserving the close coupling across modalities through a shared manifold. We notice that interesting structures appear in the learned manifold, where various manipulation properties and their possible combination with objects are organically summarized. To demonstrate the advantage of our proposed method in connecting force and vision, we evaluate our method on the two proposed tasks using our dataset. Quantitative and qualitative results are provided to show benefits of our proposed method. Additionally, we also visualize our learned manifold to better understand the latent structure captured by the model. These experiments show that our method outperforms previous multi-modal representation works. Our contributions are summarized as:

- We collect a cross-modal dataset of force and vision on real-world articulated object bodies to study hand-object manipulation
- We propose a manifold learning-based method to study the connection between the two modalities
- We conduct extensive experiments to show the advantages of our proposed dataset and method

## 2 RELATED WORK

**Shared Embedding Space** Humans interpret our visual environment through a variety of sensory modes. This notion has inspired many efforts to explore the learning of shared embeddings across diverse domains such as words (Devlin et al., 2018), images (Frome et al., 2013), audio and video (Aytar et al., 2016; Chen et al., 2023a; 2022), and text and visual data (Radford et al., 2021; Miech et al., 2019; Fang et al., 2021; Li et al., 2023). Recent efforts in this field have devoted to matching several different sensory modalities (Girdhar et al., 2023). These advancements have been enabled by the availability of large-scale, paired cross-domain datasets. However, such datasets are yet to be developed for vision and force fields. To this end, we collect a dataset of paired force fields and videos on articulated object manipulation sequences.

**Multi-Sensory Data and Learning** To equip robots with the diverse sensory capabilities, a variety of force, haptic, and tactile sensors have been innovated over the years, such as tactile (Dong et al., 2017; Yuan et al., 2017b), haptics (Lederman & Klatzky, 1987; 2009), force (Cutkosky et al., 2008) sensors. The GelSight tactile sensor has enabled many vision and robotic applications (Gao et al., 2021; Yang et al., 2022; Li et al., 2019; Yuan et al., 2017a). These GelSight tactile sensors directly produce very local geometry texture properties on a small region of on the object. Different from these prior works, the glove sensor (Sundaram et al., 2019a; Luo et al., 2021b) we work with is a full-hand force pressure sensor that directly measure the normal force maps. Existing works that maps between GelSight tactile images and RGB images (Gao et al., 2021; Yang et al., 2022; Li et al., 2019) is difficult to directly employ in our setting. The mapping between Gelsight tactile images and RGB images is more densely coupled with one-to-one mapping, because every object can have a unique surface texture property. However, the mapping between force maps and RGB images is loosely coupled many-to-many mapping, where the same object can be manipulated differently with different forces. Previous works (Zhang et al., 2021; Luo et al., 2021a) has utilized the force-pressure tactile sensors to retrieve and predict other sparse data modalities, such as motion trajectory and human pose. Different from these prior effort, we aim to map between sparse full hand force pressure maps and dense pixel sequences.

**Manifold Learning** To discover the underlying non-linear, low-dimensional subspace in which the high-dimensional input signals exist, many efforts have been devoted to learning manifold from complex sensory data (Tenenbaum et al., 2011; 2000; van der Maaten & Hinton, 2008; Donoho & Grimes, 2003; Coifman & Lafon, 2006; Belkin & Niyogi, 2003; Schölkopf et al., 1998). (Tenenbaum et al., 2011) proposes to use the geodesic distance between similar points, and (Chen et al., 2023b) proposes to use compressed latent space to achieve model reduction on physics similiation problems. Recently, many efforts (Jia et al., 2015; Mishne et al., 2015; Pai et al., 2018) have proposed the integration of traditional manifold learning algorithms with deep networks by training autoencoders to align latents with classical manifold learning embedding spaces. In this work, we aim to discover the underlying low-dimensional structure of several different sensory modalities. By enforcing the geodesic distance on dfferent sensory manifold, we allow the underlying low-dimensional structure of various sensory information to arise. Different from previous works on manifold learning and multi-modal learning, we introduce geodesic distance arithmetic on latent manifold to structure the manifold of complex sensory data in meaningful ways as supported by other data modalities.

## 3 DATASET

We focus on common articulated object bodies for our dataset. These are object and tools prevalent in our daily life. Human hand exerts complex forces on functional parts of the object to operate them in a desired way. By studying how different types of human hand haptic forces can result in different kinds of articulations, we hope to inspire interesting applications for autonomous systems to operate and manipulate these tools.

**Data Collection Setup** To collect the dataset, we leverage synchronized tactile glove sensor (Sundaram et al., 2019b) and webcam video recorder, and we follow the data recording scheme inspired by (DelPreto et al., 2022). A person wearing the tactile sensor glove to conduct various manipulations, meanwhile, we use a web cam to record the entire on hand-object interaction environment. Additionally, we also take static photos of the canonical state of all object instances. We show data collection setup in Figure. 6 in Appendix.

Our full-hand haptic force sensors are constructed following Sundaram et al. (Sundaram et al., 2019b), using an embroidery-based method. We obtain a $16 \times 16$ grid of pressure readouts distributed over the palm and finger regions of the gloves. We further calibrate the tactile sensory readings from the glove using a UR5e robotic arm to map tactile readings to force pressure level in Newton. The haptic force data is recorded in the format of a 2D heatmap on the hand, as shown in Figure. 6 in Appendix. This data format allows us to leverage commonly used image-processing network backbones (He et al., 2015) to extract the relevant force information.

**Dataset Statistics** In total, we collected 150,000 frames of synchronized haptic force and video RGB sequences. The sequence data spans 89 real-world articulated object instances, including scissors, sprayers, staplers, and clips. For each object instance, we conduct several possible manipulations. Our training set contains 81 objects spanning 4 different categories, with paired tactile and video recordings of manipulation, containing 123,561 frames of data. Our test set is constructed with 3 different subset, 1) in-distribution instances, 2) in-category novel-instances, 3) out-of-category novel instances. Specifically, the in-distribution test-set is constructed with withheld sequences of the same 81 objects seen during training, containing 10,000 frames of data. The in-category test-set contains withheld 4 objects unseen during training, containing around 6,000 frames of data. The out-of-category test set contains 4 object instances of novel category unseen during training, which contain 5,000 frames of data. In total, the test set contains: 21,000 frames of data. We group frames into small sequences for training and testing, and each sequence contains 10 frames.

## 4 VISUAL-TACTILE MANIFOLD LEARNING

In our work, we are interested in learning the cross-modal structure between RGB manipulation videos $x$, tactile signal sequence $h$. and static images of objects $c$.

Given a video sequence $x_i \in \mathbb{R}^{10 \times 64 \times 64 \times 3}$, we would like to infer the corresponding tactile sequence observation $h_i \in \mathbb{R}^{10 \times 16 \times 16 \times 1}$ and what object is being manipulated in the image $c_i \in \mathbb{R}^{64 \times 64 \times 3}$, and likewise given a tactile reading $h_i$, infer the corresponding possible visual inputs $x_i, c_i$. The chief difficulty is the dramatic signal disparity between these modalities. We present a method to capture and generate the shared structure between signals by embedding them into a shared latent space, with individual signals being decode from the latent using a neural field. We further propose to leverage manifold projection to encourage the desired structure to arise from the learned manifolds.

### 4.1 LEARNING A SHARED CROSS-MODAL VISUAL-TACTILE MANIFOLD

Given a training set of paired video, tactile, and image tuples $\mathcal{I} = \{x_i, h_i, c_i\}_{i=1}^N$, we would like to learn and discover the shared structure between different signals. A chief difficulty is the dramatically different nature between tactile and visual signals, with visual videos being significantly higher resolution and complex than corresponding tactile signals.

Existing methods, such as TouchandGo Yang et al. (2022), ObjectFolder Gao et al. (2021), VisGel Li et al. (2019), etc., employ encoder-decoder techniques where the latent codes are constructed and obtained from encoding a specific modality of information. To effectively capture structure across such dramatically different signals, we propose to compactly represent each signal in a shared latent space, where tactile, video, and image signals with three separate neural fields. Specifically, we use an auto-decoder technique, using different branches of neural field decoders to construct different modalities of information (tactile/vision). In this way, we allow gradients from different modalities

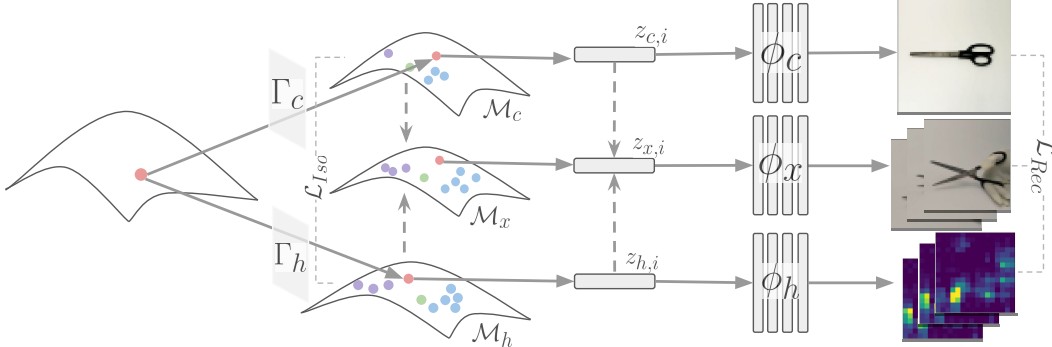

Figure 1: **Method Overview.** Our approach embeds image, tactile, and video information into a shared latent manifold. This manifold is further decomposed into submanifolds for each signal.

of information to simultaneously update and optimize the latent embedding. We randomly initialize our shared latent embedding.As shown in Figure. 1, the latent embedding on the leftmost side is the learned shared manifold. During training, we sample a latent code $z_i$ to decode to different modalities of signals, $x_i$, $h_i$, and $c_i$ shown on the right side.

Formally, given either visual or tactile signal, we represent it efficiently as a function $\Phi$ which maps each coordinates $\boldsymbol{u} \in \mathbb{R}^l$ in either image or tactile signals to the value $v$ of the feature coordinate at that dimension.

$$\Phi : \mathbb{R}^l \to \mathbb{R}, \quad \boldsymbol{u} \to \Phi(x) = v \tag{1}$$

Such a formulation enables us to represent any signal, such as an image, shape, video sequences, or waveform as a continuous function $\Phi$. We use MLPs to represent the neural field $\phi$. Only the input dimension of $\Phi$ is varied, depending on the dimensionality of the underlying signal – this allows us to employ the same architectural choices across signal types.

## 4.2 CROSS-MODAL MANIFOLD PROJECTION

In order to effectively capture shared structure across a image, tactile and visual signals we would like our shared cross-model visual-tactile *manifold* to not only smoothly interpolate and project between different instances of signals to each other, but also capture the local structure in each individual signal modality. To enforce this local structure, we propose a *manifold projection layer*, where we linearly project our global manifold space to local submanifolds for each signal. We then adopt existing manifold learning losses to ensure that each submanifold captures a space that is locally consistent and metric.

Formally, the manifold projection layer $\Gamma_k : \mathcal{M} \to \mathcal{M}_k, k \in \{x, h, c\}$ takes in the signal agnostic manifold to project to a compressed lower dimensional submanifold that reflects one aspect of a signal modality. As shown in Fig. 1, the sampled latent code (red dot in the shared manifold) is projected through $\Gamma_c$ and $\Gamma_h$ to the signal-specific manifold $\mathcal{M}_c$ and $\mathcal{M}_h$. To encourage differentiability and explainability, we choose $\Gamma_k$ to be a linear projection $\Gamma_k \in \mathbb{R}^{S \times T}$, where $S$ is the dimension of the original signal agnostic manifold and $T$ is the dimension of the sub-manifold. Each data modality $k \in \mathcal{I}$ is decoded from the submanifold $\mathcal{M}_k = \Gamma_k(\mathcal{M})$ and we enforce that any two samples sampled from the signal agnostic manifold $\{z_i, z_j\} \subseteq \mathcal{M}$ respects the distance between the samples $\mathcal{I}_i, \mathcal{I}_j$, **and** that the same two samples sampled from the submanifold $\{z_{k,i}, z_{k,j}\} \subseteq \mathcal{M}_k$ should respect the distance between the samples for the $k$-th modality of signal $\mathcal{I}_{k,i}, \mathcal{I}_{k,j}$.

For cases where some signal modalities can be very expressive and contain rich and entangled information, this type of submanifold projection method can help disentanglement and encourage various structures to arise from the sub-manifold focusing only on the specific signal being described. For example, in our test case with video $x$ and tactile $h$ sequence data, the video $x$ sequences contain rich information about the manipulation and objects being manipulated, whereas the tactile $h$ sequence data is sparse and reflects mostly the hand manipulation and less about the object geometry. Using a natural language analogy, as video data can be described using simple sentences, the tactile information focuses on the verb of the sentence, and the noun can be captured by a static image of the canonical shape. Through a submanifold projection, we can encourage one submanifold $\mathcal{M}_h = \Gamma_h(\mathcal{M})$ to be structured around the tactile modality and focus on manipulation. Similarly, we can leverage the submanifold projection $\mathcal{M}_c = \Gamma_c(\mathcal{M})$ to

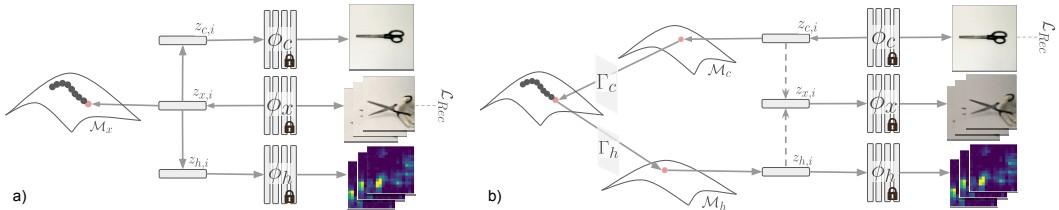

Figure 2: **Test-Time Manifold Manipulation.** Our learned manifold can be used to retrieve possible cross-modal completions of a relevant signal, including those of novel unseen test objects. a) shows the process of cross-modal prediction, and b) shows the process of inference on novel shape instances.

focus on object geometry concepts in canonical object images. We compose video latents $z_{x,i}$ by combining latents of manipulation $z_{h,i}$ and the latents of canonical shapes $z_{c,i}$, As shown in the middle branch in Figure. 1. Specifically, the video projection layer $\Gamma_x$ is constructed by concatenating the image projection $z_{c,i} = \Gamma_c(z_i), \Gamma_c \in \mathbb{R}^{256 \times 256}$ layer and tactile projection layer $z_{h,i} = \Gamma_h(z_i), \Gamma_h \in \mathbb{R}^{256 \times 16}$ $z_{x,i} = \Gamma_x(z_i) = \{\Gamma_c; \Gamma_h\}(z_i)$, where $\Gamma_x = \{\Gamma_c; \Gamma_h\} \in \mathbb{R}^{256 \times 272}$, or equivalently, $z_{x,i} = \{\Gamma_c(z_i); \Gamma_h(z_i)\} = \{z_{c,i}; z_{h,i}\}$. This way of enforcing the compositional property in manifold space can help disentangle rich and expressive into more elementary concepts.

To further encourage structures of elementary concepts to arise from the submanifolds, we would like to distill the dominant and descriptive information from each sensory modality onto their submanifolds. Inspired by Chen et al. (Chen et al., 2023b), we choose to project the tactile submanifold $\mathcal{M}_h$ to extreme low dimensions to constrain the neural field to focus on the dominant and most descriptive information. For our tactile submanifold $\mathcal{M}_t$ we choose dimension=16.

To ensure our learned submanifolds follow the desired properties of data completeness and perceptual consistency, we use the following loss functions, as inspired by (Du et al., 2021). First, to ensure reconstruction accuracy, we train each auto-decoder branch using the reconstruction loss.

$$\mathcal{L}_{\text{Rec}} = \|\phi_c(z_{c,i}) - c_i\|^2 + \|\phi_h(z_{h,i}) - h_i\|^2 + \|\phi_x(z_{x,i}) - x_i\|^2 \tag{2}$$

Additionally, to ensure that our learned manifold achieves perceptual consistency, where similar signals are closer in the underlying manifold. We employ the manifold isometry loss:

$$\mathcal{L}_{\text{Iso}} = \|\alpha * \|(z_i - z_j)\| - \|\mathcal{I}_i - \mathcal{I}_j\|\|. \tag{3}$$

$$= (d_z(z_{c,i}, z_{c,j}) - d_{\mathcal{I}}(c_i, c_j)) + (d_z(z_{h,i}, z_{h,j}) - d_{\mathcal{I}}(h_i, h_j)) + (d_z(z_{x,i}, z_{x,j}) - d_{\mathcal{I}}(x_i, x_j)) \tag{4}$$

$$= (\|\|(z_{c,i}) - (z_{c,j})\| - \|c_i - c_j\|\|) + (\|\|(z_{x,i}) - (z_{x,j})\| - \|x_i - x_j\|\|) + (\|\|(z_{h,i}) - (z_{h,j})\| - \|h_i - h_j\|\|) \tag{5}$$

In this way, we enforce the distance between latents $z_i$ and $z_j$ in our manifold $\mathcal{M}$ to be locally isometric to the MSE distance of the corresponding samples $\mathcal{I}_i$ and $\mathcal{I}_j$. Our loss function is constructed using both terms: $\mathcal{L} = \mathcal{L}_{\text{Iso}} + \mathcal{L}_{\text{Rec}}$.

### 4.3 MANIPULATING THE TACTILE-VISUAL MANIFOLD

Given our shared tactile-visual manifold, we next describe ways to manipulate and leverage the learned tactile-visual manifold to map between visual and tactile signal.

**Cross-Modal Signal Retrieval**. During test-time, given a seen tactile $h_i$ or visual $x_i, c_i$ signal, we would like to retrieve the corresponding signals of the other modalities. As shown in Figure. 2, a) we leverage test-time latent code optimization to achieve cross-modal retrieval. For example, given the RGB visual sequence $x_i$, and we would like to see the corresponding tactile sequence $h_i$. We first start at a random location on the learned manifold $\hat{z}_{x,i}$, which corresponds to an rgb video signal $\phi_x(\hat{z}_{x,i})$. We use MSE as loss signal to optimize for the latent, $\arg\min_{\hat{z}_{x,i}} \|x_i - \phi_z(\hat{z}_{x,i})\|^2$, by fixing weight of the networks $\phi_c, \phi_x, \phi_h$. The optimized latent $\hat{z}_{x,i}^*$ can be directly decomposed into latent vector for tactile sequence $\hat{z}_{h,i}$ and $\hat{z}_{c,i}$. We can use the projected latent to decode the desired tactile sequence $\hat{h}_i = \phi(\hat{z}_{h,i})$.

Similarly, given a tactile sequence $h_i$, we can retrieve the corresponding video sequence $x_i$ by optimizing latent $\hat{z}_{h,i}$ to project back to the canonical image space $\mathcal{M}_c = \Gamma_c(\Gamma_h^\dagger(\mathcal{M}_h))$ through the signal agnostic manifold to obtain the corresponding canonical image vector $\hat{z}_{c,i}$. We use † to denote

the reverse direction of the original projection. With the two vectors combined $\hat{z}_{x,i} = [\hat{z}_{c,i}; \hat{z}_{h,i}]$, we can decode into the desired sequence of visual signal, $\hat{x}_i = \phi_x(\hat{z}_{x,i})$.

One other advantage of our proposed method is varitational inference. Without any modification to the it can be used to predict variations of different signals our framework learns the joint manifold of visual and tactile signals. This allows us to handle multimodal settings, as each possible pairing is in the manifold. This can be extracted by running test-time optimization with different latent initializations. By initializing different latents for test-time optimization, we can allow variational inference for fixed given sequence. We show an example in Section A.3 in Appendix.

**Cross-Modal Prediction for Unseen instances** Another advantage of our proposed method is its ability to handle novel object instances. As shown in Figure.2 b), given an image of an unseen object instance $j$, our proposed method can be used to predict possible manipulation force and video sequences. We can interpolate the shape manifold space to optimize for a shape latent code $\hat{z}_{c,j}$ that matches the image signal of the object instance. Various method exists to infer manipulation sequences $\hat{h}_j, \hat{x}_j$. If a desired manipulation is given through a tactile sequence $h_j$, we can optimize for the tactile latent code $\hat{z}_{h,j}$. Alternatively, we can infer the possible manipulations by projecting the shape latent code onto the manipulation manifold $\hat{z}_{h,j} = \Gamma_h(\Gamma_c^\dagger(\hat{z}_{c,j}))$. With the desired tactile manifold vector $\hat{z}_{h,j}$, we can decode tactile manipulation sequences from $\hat{h}_j = \phi_h(\hat{z}_{h,j})$, and video sequence from the latent vectors $\hat{x}_j = \phi_x([\hat{z}_{c,j}; \hat{z}_{h,j}])$. When given a video or tactile manipulation sequence of a novel instance, our proposed method can predict the other modality by interpolating and optimizing latent $\hat{z}_{k,j}$ on the latent manifold.

## 5 EXPERIMENT

In this section, we show several experiments to answer the following questions:

- How does our proposed method perform compare to existing cross-modal retrieval methods?
- Can our proposed method generalize manipulation seen during training to novel object instances?
- Does our learned manifolds exhibit meaningful latent structure of each signal modality?
- How is our dataset and method used in robotic and vision applications? (Appendix Sec. A.2)
- Does our proposed method enable variational inference? (Appendix Sec. A.3)

Table 1: **Quantitative comparison on cross-modal prediction.** Our approach substantially outperforms baselines across each test on cross-model signal undersanding.

| Method | cross-modal retrieval | | generalize to new object | | generalize to new category | |
|---|---|---|---|---|---|---|
| | $x \to h$ | $h \to x(h, c \to x)$ | $x \to h$ | $h \to x(h, c \to x)$ | $x \to h$ | $h \to x(h, c \to x)$ |
| Visgel | 0.0022 | 0.0038 | 0.0024 | 0.0139 | 0.0024 | 0.0176 |
| TouchGo | 0.0027 | 0.0262 | 0.0028 | 0.0228 | 0.0025 | 0.0224 |
| ObjectFolder | 0.0030 | 0.0340 | 0.0034 | 0.0236 | 0.0026 | 0.0216 |
| GEM | 0.0022 | 0.0194 | 0.0023 | 0.0106 | 0.0021 | 0.0160 |
| Ours | 0.0019 | 0.0174 (0.00208) | 0.0020 | 0.0097 (0.00710) | 0.0020 | 0.0118 (0.01038) |

### 5.1 BASELINES

We compare our proposed approach with several multi-sensory retrieval methods:

- VisGel (Li et al., 2019): Li et al. proposes to a conditional-GAN based model to synthesize GelSight tactile images from input video sequences. The model takes in a video sequence and conditioned on a RGB reference frame and a GelSight reference frame to predict the tactile sequence. We compare with the proposed approach by using the canonical shape image as reference to conduct cross-modal prediction on our force-visual data.
- ObjectFolder (Gao et al., 2021): Gao et al. proposes to use triplet loss to associate the sensory modalities of RGB images, GelSight images, and sound. Inspired by them, we adapt the method as a baseline on our haptic force and vision data to use a triplet loss to anchor the force sequences, video sequences, and canonical shape images.
- TouchAndGo (Yang et al., 2022): Inspired by previous works (Li et al., 2022; Tian et al., 2020), Yang et al. proposes visuo-tactile CMC model to align RGB images that GelSight tactile images. We compare with their proposed approach by adapting it to work on our visuo-force data.

- GEM (Du et al., 2021): Du et al. proposes the signal-agnostic manifold learning based method to represent and translate different sensory modalities. GEM (Du et al., 2021) was originally proposed for cross-modal visual-audio retrieval. We modify it to work with our haptic force and vision data.

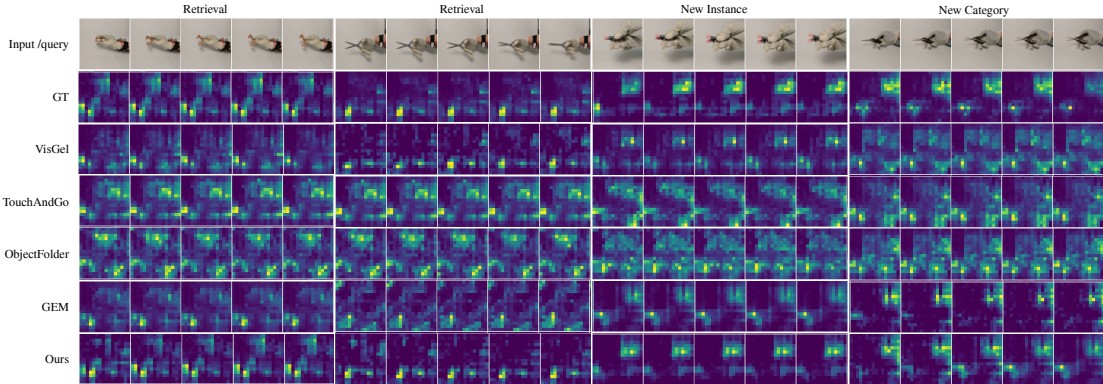

Figure 3: **Qualitative Results for Video-to-Tactile Prediction.** A comparison of $x \to h$ comparing our proposed method and existing methods on cross-modal retrieval and generalization to new category and object instances. We show 5 frames in each example. Yellow regions corresponds to high pressure areas. Our approach is able to outperform baselines.

## 5.2 EVALUATION METRIC AND QUANTITATIVE RESULTS

We use L2 distance between the predicted signal and ground truth signal to evaluate the performance of various methods, $\|\hat{k}_i - k_i^{gt}\|^2, k \in \{c, x, h\}$ . We show the quantitative performance in per-pixel difference for our proposed method and existing methods in Table. 1.

To compare with existing cross-modal prediction methods, we test on with several different task settings. First, following previous works (Li et al., 2019; Yang et al., 2022), and evaluate the cross-modal retrieval capabilities of different methods. This evaluation involves a subset of training object instances and manipulation sequences, as depicted in the *cross-modal retrieval* section of Table 1. Additionally, we would like to understand if the proposed method is able to extend its learned manipulation insights to novel object instances in a seen object category. We show the cross-modal prediction performance in the context of unseen object instances of the training object category in *generalize to new object* section. Finally, we also conduct a stress test to see the ability of various method in generalizing to novel instances in unseen object category to understand if the proposed, as shown in *generalize to new category* section.

We can see from Table. 1, our proposed method outperforms existing methods on both the cross-modal retrieval task and the tasks of cross-modal generalization to new object instances and category. The task difficulty increases from retrieval to generalization. All methods exhibits performance decrease when tested on novel instances and stress-tested on novel category.

We observe a performance gap across all methods between inferring the tactile modality from videos $x \to h$ and predicting video sequences from tactile signals $h \to x$. The drop in performance comes from the inherent difference between the two signal modalities. Videos contain rich and dense information about both the manipulation and the object being manipulated, whereas tactile signals only capture the miniature portion of object through the force field actuated on small areas of our hand. In this sense, predicting videos sequences from only tactile signals $h \to x$ is inherently a more difficult task. As Table. 1 shows, when non-sufficient information is given, it is easy for test-time optimization based methods, such as GEM (Du et al., 2021) and our proposed approach, to stuck in local minima, whereas conditioning-based approach, such as VisGel (Li et al., 2019), can leverage the close coupling of conditioning vector $z_h$ and output signal $x$ to achieve better retrieval accuracy when predicting video from tactile. However, due to the same reason of the close coupling of the conditioning vector and output, conditioning-based approaches are limited in generalizability to new object instances and categories. Our proposed approach, on the other hand, encourage discovery of structured information embedded in each signal modality, and, therefore, is able to transfer manipulation signals learned through seen data to novel instances and object categories. Other methods that leverage contrastive losses, such as TouchAndGo (Yang et al., 2022) and ObjectFolder (Gao et al., 2021), encourages loose coupling of latent vectors of different modalites

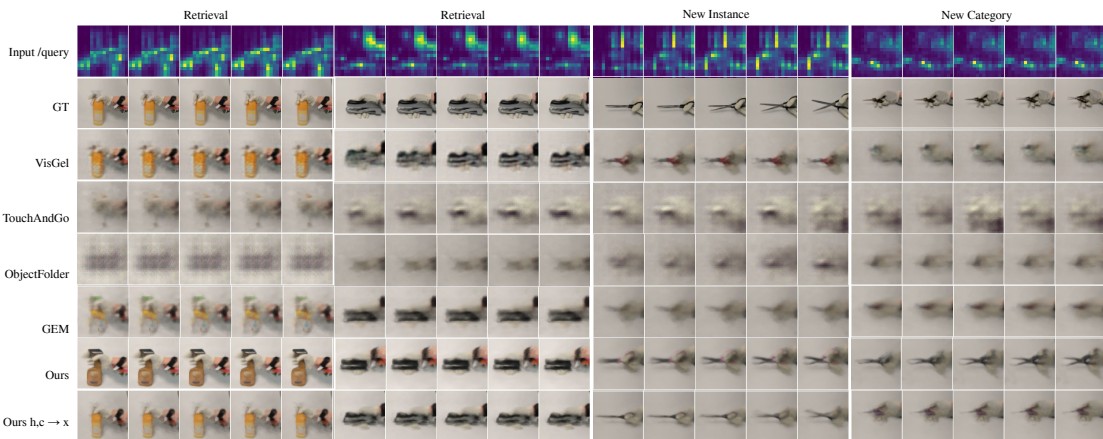

Figure 4: **Qualitative Results for Tactile-to-Video Prediction** We illustrate $h \rightarrow x$ results comparing our proposed method and existing methods on cross-modal retrieval and generalization to new category and object instances. We show 5 frames of sequence data in each example.

$z_h, z_x$. In our context of extreme scale mismatch, these metric-learning based methods fail to produce meaningful cross-modal prediction and result in mode collapse.

In Table. 1, we also show additional results of our method leveraging manifold composition capability. Instead of only leveraging tactile signals $h$, we compose information of both object instance $c$ and manipulation type $h$ to predict the video of object manipulation $x$, as shown in $h, c \rightarrow x$. We can see from the table that our proposed approach is able to leverage the extra bit of information $c$ to significantly improve the accuracy of our video prediction. This demonstrates that our proposed approach is able to disentangle object concept and manipulation concepts leveraging different signal modalities, and compose these elementary concepts into meaningful visual sequences.

## 5.3 CROSS-MODAL PREDICTION QUALITATIVE RESULTS

We show the qualitative results of various method on the task of cross-modal retrieval and generalization to new object instances and category in Figure. 3 and Figure. 4 .

**Video to Tactile** $x \rightarrow h$ Predicting tactile manipulation signal is a relative easier task as compared to tactile-to-video task. We can see from Figure. 3 that both our proposed approach and Visgel (Li et al., 2019) is able to accurately retrieve tactile sequences given video sequences seen during training. In generalizing to novel instances and category, our proposed method is able to capture the manipulation in the video to and accurately predict tactile signal sequences, whereas other existing method fail to predict meaningful tactile sequences for unseen objects.

**Tactile to Video** $h \rightarrow x$ We can see from Figure. 3 that when directly predicting videos from the tactile information, most existing approaches, including TouchAndGo (Yang et al., 2022), ObjectFolder (Gao et al., 2021), suffers from mode collapse. As the query manipulation tactile signal is extremely sparse, contrastive losses such as CMC loss used in TouchAndGo or Triplet loss used in ObjectFolder are unable to retrieval meaningful video latent code $z_x$ when the anchor latent vector $z_h$ are of extreme resemblance in metric space. VisGel (Li et al., 2019) uses a conditional-GAN and retrieves seen video sequences accurately, but it has limited generalizability to novel instances and categories. GEM (Du et al., 2021) leverage a shared latent manifold to predict one modality from the other. However, as different modality of signals are of different scale, the latent vector shared by different modality of signal become dominated by the modality with more information and weakens the model's ability to be queried by the sparser modality. As a result, the retrieved video sequences by GEM exhibits a smoothing effect. However, when only tactile information is given, our proposed approach retrieves sharper videos, but the subject shape instance is often wrong. For example, with the new black scissor instance, our proposed method predicts a manipulation sequence of a red scissor. Tactile force signal is not informative of the color of the object. When we add the additional information with a canonical object image $h, c \rightarrow x$, our proposed method is able to correctly predict the manipulation sequence of the desired object.

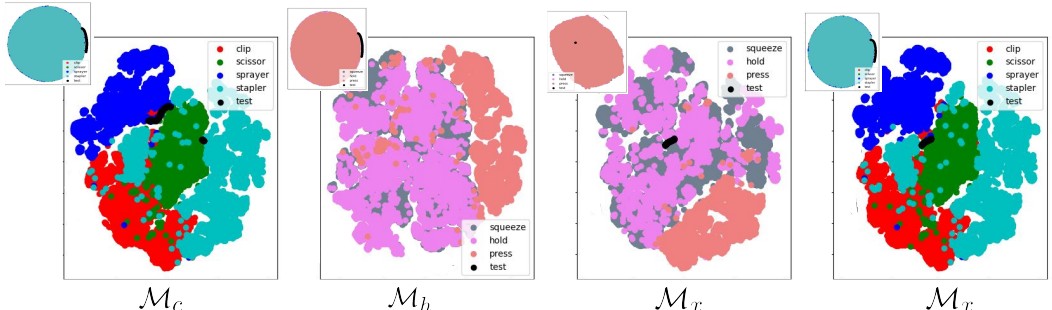

Figure 5: **TSNE Visualization of Learned Latent Manifold.** We visualize the manifold structure captured by our approach each submanifold and across the full manifold. On the top left corner, we show the initialization of each manifold. $\mathcal{M}_c$, $\mathcal{M}_h$, and $\mathcal{M}_x$ is the shape, tactile, and video manifold respectively. We color the latent vectors by their object and manipulation category. The trajectory of optimizing a latent code on the learned manifold during test-time is shown in black.

### 5.4 LATENT INTERPRETABILITY

We would like to understand whether our proposed method can encourage meaningful structures to naturally arise on the learned manifold without labeled data. We visualize our learned latent manifold through a T-SNE embedding as shown in Figure. 5. We can see from the figure that randomly initialized latent embedding are trained to accurately capture semantic concepts such as object category and manipulation type without any explicit label supervision.

We also observe that manipulation manifold $\mathcal{M}_h$ trained only with tactile sequence signal shows an interesting structure, where the manipulation "press" is clustered and away from the other manipulations of "hold" and "squeeze". We reason that the manipulation action of pressing is different from the other manipulation actions in that the actions of pressing activates a contiguous area of the hand and is one directional, whereas the manipulation actions of "squeeze" and "hold" create force closure through opposing directional force actuated by multiple areas of the hand. In this sense, the actions of "squeeze" and "hold" share more similarities.

As the video sequence manifold $\mathcal{M}_x$ is obtained through compositing $\mathcal{M}_c$ and $\mathcal{M}_h$, it now shares the interesting property of both manifolds. Objects of similar manipulation and visual properties are clustered together, such as the staplers and scissors that are both two rigid body parts connected through a rotational axis. Tools that are enabled by similar manipulation action are now also clustered together, such as the action "press" for the objects of "clip" and "stapler".

**Remark.** With the above experiments, we have demonstrated that our proposed approach outperforms exhisting methods in cross-modal prediction, and is able to encourage meaningful latent structure to arise from different modality of signals. We do want to note several limitations of our work, we primarily focus on concept discovery and understanding manipulation and object, our decoding neural field is a simple MLP and more advanced neural field architectures can be employed and combined to reconstruct high frequency details and improve the reconstruction quality, which we leave for future work. Additionally, we also show several ablation studies(Sec. A.4), robotic applications (Sec. A.2), variational inference (Sec A.3), and object classifcation experiments (Sec. A.2) in the Appendix.

## 6 CONCLUSION AND FUTURE WORK

In summary, we take an manipulation perspecitve to gain insights into objects and tools. Our primary focus is to explore the intricate connection between manipulation and objects. To this end, we curate a multi-modal dataset encompassing both visual and tactile information that contains paired full-hand force pressure maps and manipulation videos. Furthermore, we introduce an innovative technique designed to uncover a cross-modal latent manifold. This manifold enables cross-modal predictions and the revelation of hidden structures within distinct data modalities. We present a series of extensive experiments to demonstrate that our proposed method learns manifolds of meaningful latent structure that captures the principled information in each data modality. Our proposed approach outperforms existing cross-modal methods and exhibits ability to generalize learned manipulation to novel object instances.

## 7 ACKNOWLEDGEMENT

We thank Wistron and GIST for their research grant for supporting our project.

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

## A  SUPPLEMENTARY

### A.1  DATA COLLECTION SETUP AND TACTILE-HAND VISUALIZATION

Figure. 6 shows our data collection setup and tactile-hand visualization to facilitate understanding of the hand-tactile pressure map signal. **Additional Visualization and Data Overview** We add

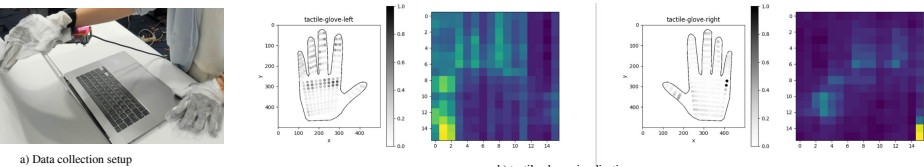

a) Data collection setup  b) tactile glove visualization

Figure 6: Data Collection Setup and Tactile-Hand Visualization

additional visualization to illustrate our data in Fig. 7. Visualization of motion sequences can be found on our supplementary website. **Background on Glove** The glove is made using piezoresistive

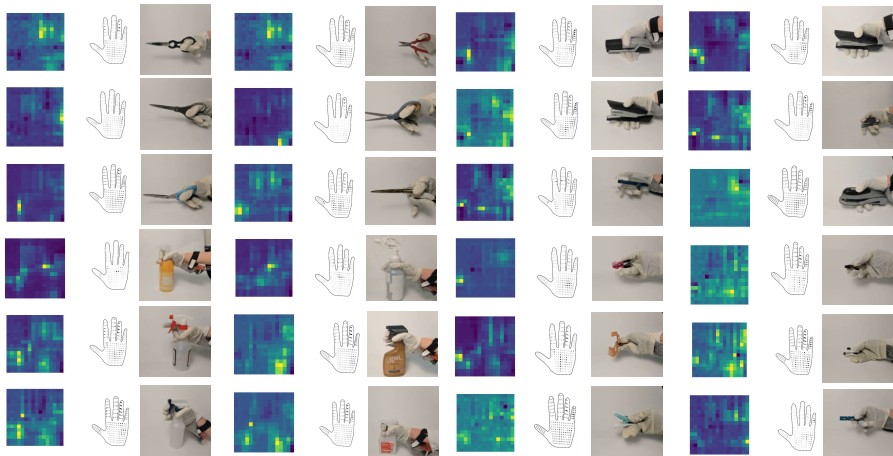

Figure 7: Data Visualization. We show a demo of our data with 12 samples from our 89 objects.

film. The working principle of this material is when the material stretches, it causes the deformation of carbon nanotubes, which then induces the relative electrical resistance changes and thus difference in voltage readouts. The material in general is sturdy and the material property does not change much with wear and tear.

**Glove Calibration.** The glove only needs to be calibrated once to convert the sensor readouts in voltages to force magnitudes in Newton before use. We show the process of data calibration in Fig. 8.

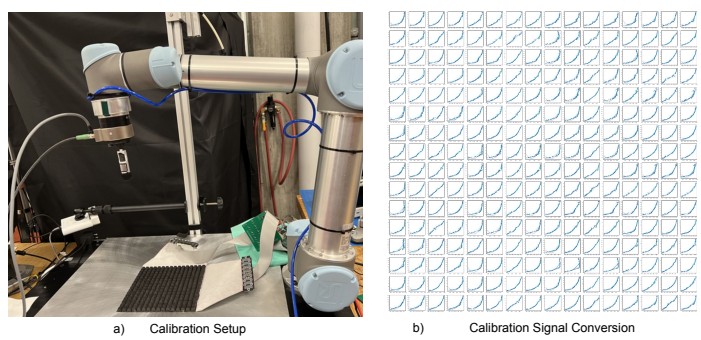

a)  Calibration Setup  b)  Calibration Signal Conversion

Figure 8: Sensor Calibration. The left shows the sensor calibration setup before the embroidery of the insulation layer. The right shows the collected calibration data from 0 to 50N of continuous force applied to each of the $16 \times 16$ voltage readout positions.

## A.2 DOWNSTREAM TASKS

To motivate downstream robotic applications, we show two experiments inspired by robotic perception applications. The high-level intuition is that if our learned embedding space trained with human-collected data can sufficiently represent data collected by robotic arms, we can transfer knowledge from human demonstration to robotic tasks. First, we collected some data using a robotic hand to show the cross-modal prediction results leveraging the pretrained embedding with human data. Additionally, we would like to see how much knowledge is transferrable from human demonstrations to data to a robotic hand. To achieve this goal, we train a simple classifier using our trained embedding space with human data to enable manipulation and object classification through tactile sequences. While we only show two applications here, other downstream tasks can be enabled by our proposed method and dataset.

**Robotics inspired application.** We recognize that different robotic hands might induce different levels of domain gap. To minimize the domain gap, we use a robotic motor to drive a tenden-driven OpenBionics hand, and record a tactile sequence shown below Fig. 9 a) and b). We use test-time optimization approach to find a latent code in the shared embedding space to decode into other modalities. We notice that because the robotic hand is non-conformable, the collected sequences of data are of inherent domain gap and thus result in different possible predictions that equally fit the optimization energy.

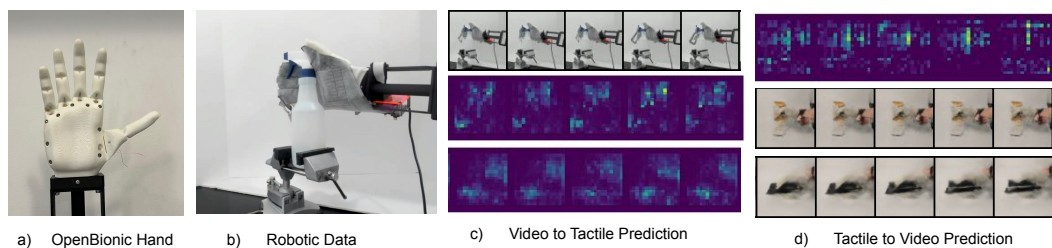

a)   OpenBionic Hand          b)   Robotic Data          c)   Video to Tactile Prediction          d)   Tactile to Video Prediction

Figure 9: Robotic application using shared manifold learning trained with human collected data. a) and b) show the task and data collection setup. c) d) shows the cross-modal prediction results of only using the video or the tactile sequence to infer the other modality of signals. The top row shows the query signal, and the bottom two rows showcase different predictions made by our method.

**Object classification with shared embedding .** One other potential application is to help robotic tasks in low-light settings, by equipping robotic hands with tactile sensors. We can leverage our pretrained cross-modal manifold to recognize objects and even operate objects when vision is impaired. With our trained shared latent space we train a classifier to classify the corresponding object category and manipulation type from the signal sequence. We withhold some latent codes from the classifier to test the baseline performance on human data. We use test-time optimization to obtain a latent code in the shared embedding space using the robotic tactile data. We show the performance in Table. 2 below.

| test data type | manipulation | object |
|---|---|---|
| human test data | 0.96 | 0.97 |
| robotic tactile data | 0.86 | 0.48 |

Table 2: Object classification of human and robotic data using shared embedding space

We notice that human manipulation and robotic manipulation share a significant amount of transferrable and common knowledge. In that a classifier trained using the shared latent code that has only seen human data and also be used to classify robotic manipulation data. On the other hand, the non-conformability of the robotic hand significantly restricts the information on the details of the object geometry in the tactile signal and thus limits the accuracy of classifying objects using robotic tactile data.

While these are simple robotic applications of our proposed dataset and method, we note here that other more advanced robotic applications are possible. We focus on learning a joint representation for different modalities of signals and leave more challenging robotic tasks as future work.

## A.3 MULTI-MODAL PREDICTION

We randomly initialize three different latent codes to start our test-time optimization for a given tactile video sequence. We show that we are able to recover different cross-modal sequences with different initializations, shown in Figure. 10. We want to note that it is possible that different latent initializations converge to the same mode of prediction when there exists a global optima, or when two latent codes are initialized around the same neighborhood of a local minima.

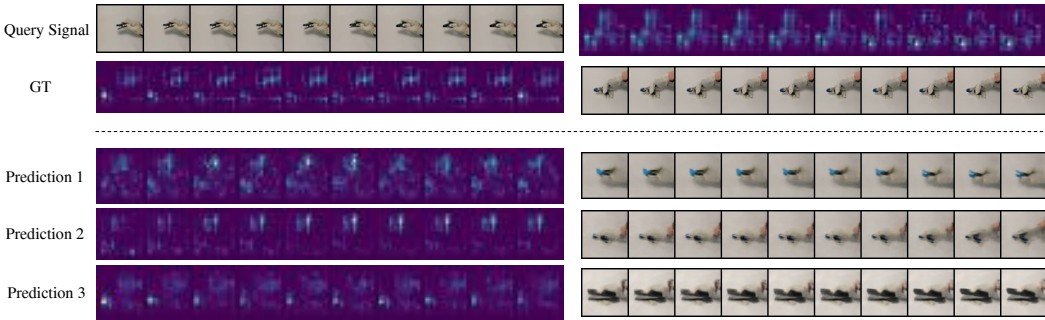

Figure 10: MultiModal Prediction via variational initialization test-time optimization.

## A.4 ABLATION EXPERIMENTS

Table 3: Ablation Study

|  | video to tactile | tactile to video |
|---|---|---|
| first frame inference | 0.0028 | 0.0247 |
| w/out projection layer | 0.0022 | 0.0194 |
| dim $z_h$= 256 | 0.0018 | 0.0212 |
| dim $z_h$= 128 | 0.0018 | 0.0201 |
| dim $z_h$= 64 | 0.0019 | 0.0196 |
| dim $z_h$= 5 | 0.0023 | 0.0188 |
| ours | 0.0019 | 0.0174 |

We show our ablation experiments in Table. 3. We observe that removing the projection layer results in decrease of cross-modal prediction performance. We can also observe that by varying the latent dimension of the tactile signal $z_h$, the performance for video to tactile prediction does not change too much, which means that $z_h$ is over-parameterized. Through decreasing the latent dim $z_h$, performance for tactile to video starts to increase. when latent dimension decreases, the neural field enforce the latent code to use limited space to register the same amount of information, and thus captures the principled information in sparse signals such as tactile. Our proposed method leverages latent dimension of 16, which trades off the marginal gain of accuracy reconstructing tactile signal, but captures much more principled information in the tactile space. Finally, we also show that our model can also be directly adopted to only use the first frame signal during test-time optimization, we notice a performance decrease from using the full sequence data. We believe the difference resulted from an increase of variational inference, or in other words, with less optimization signal, there are now more local minima during test-time optimization that fit the optimization objective.

## A.5 IMPLEMENTATION DETAILS

All our manifolds are randomly initialized with Gaussian distribution with $(\mu, \sigma) = (0, 1)$. Tactile manifold $\mathcal{M}_h$ is initialized to be 16 dimensional and all other manifolds are initialized to be 256 dimensional. We use MLPs to express our neural fields $\Phi$. All MLP neural fields are of three layers and 512 hidden dimension. Specifically, the shared manifold is initialized to be an 256 dimensional embedding of with $N_{train} = 123, 561$ randomly sampled from a Gaussian distribution of $(\mu, \sigma) = (0, 1)$. $\Gamma_c$ is parameterized by a $256 \times 256$ matrix, and $\Gamma_h$ is parameterized using a

$256 \times 16$ matrix. Neural fields $\phi_c, \phi_x, \phi_h$ are all 6-layer MLPs using ReLU activation and $512$ dimensional hidden-layers. We also use positional encoding for our signal-agnosticneural field with frequency coefficient set to $\omega = 30$.

## A.6 TRAINING DETAILS

All baseline methods and our proposed method are implemented using the open-source Py-torch (Paszke et al., 2019) package with CUDA 11.7 backend. We use the same hardware and training setup to train all methods mentioned in this work. We use a workstation equipped with NVIDIA Tesla V100 GPUs and 64-core AMD CPUs. We use Adam optimizer for training the baselines and our methods with learning rate initialized to be $1e - 3$; batch size is set to 64. All methods are trained to full convergence. For test-time optimization of latent code for GEM (Du et al., 2021) and our proposed approach, we use Adam optimizer to run 1000 timesteps with batchsize of 32. For 1000 timesteps, we use 56 seconds in total, 17.87 iteration/second, on average every example use 1.75 seconds.

