# OpenReview forum: "Learning to Jointly Understand Visual and Tactile Signals"
_ICLR.cc/2024/Conference — ICLR 2024 poster_

### Official Review · Reviewer_RFKG · 2023-10-23

**Soundness:** 2 fair
**Presentation:** 3 good
**Contribution:** 2 fair
**Rating:** 6
**Confidence:** 3

**Summary:**

The paper studies the problem of learning representations for modeling visual and tactile sensor data. A large multi-modal visual-tactile dataset is presented, and a straightforward pipeline is proposed for learning the data. Experiments are performed on cross-modal prediction tasks to validate the idea.

**Strengths:**

The paper introduces a substantial dataset comprising paired visual and tactile sensor data, which holds the potential for significant advancements in cross-modal research;

The paper's organization is clear and easy to follow.

**Weaknesses:**

The proposed approach has been exclusively assessed in the context of cross-modal prediction tasks, with no concrete verification of its applicability in downstream manipulation tasks;

Moreover, it is worth noting that a single video observation could potentially correspond to a wide range of tactile signals, such as variations in the force applied when touching dough. Regrettably, the study does not appear to account for the inherent multimodality in the distribution of data in this respect;

The paper lacks technical details, e.g., the learning rate, batch size, etc.

The video prediction results for new instances and new categories seem not promising.

**Questions:**

Providing additional technical details would enhance the reproducibility of the work, e.g., model architecture, training details, etc.

It would be interesting to see how the learned representations can be applied to downstream manipulation tasks, adding such results would further strengthen the paper.

---

> ### Author Response · Authors · 2023-11-20
> **Update on 1) added experiments on a robotic-manipulation-inspired task, 2) variational inference experiment, 3) details on model architecture and training setup**
>
> We thank the reviewer for the feedback. We have updated our draft with 1) added experiment on a robotic-manipulation-inspired task, 2) variational inference, 3) details on model architecture and training setup
>
> **1. The proposed approach has been exclusively assessed in the context of cross-modal prediction tasks, with no concrete verification of its applicability in downstream manipulation tasks; (Question 2) It would be interesting to see how the learned representations can be applied to downstream manipulation tasks, adding such results would further strengthen the paper.**
>
> We thank the reviewer for this question, we added a robotic-manipulation-inspired experiment in our draft in Appendix Sec. A3.
>
> To motivate downstream robotic applications, we show **two experiments** inspired by robotic perception applications. The high-level intuition is that if our learned embedding space trained with human-collected data can sufficiently represent data collected by robotic arms, we can transfer knowledge from human demonstration to robotic tasks. First, we collected some data using a robotic hand to show the cross-modal prediction results leveraging the pretrained embedding with human data. Additionally, we would like to see how much knowledge is transferrable from human demonstrations to data to a robotic hand. To achieve this goal, we train a simple classifier using our trained embedding space with human data to enable manipulation and object classification through tactile sequences. While we only show two applications here, other downstream tasks can be enabled by our proposed
> method and dataset.
>
> To minimize the domain gap between robotic hand and human hand, we use a robotic motor to drive a tenden-driven OpenBionics hand, and record a tactile sequence shown in Fig. 8 in appendix. We use a test-time optimization approach to find a latent code in the shared embedding space to decode into other modalities. We show that we can perform cross-modal inference with robotic data.
>
> Object classification with shared embedding. One other potential application is to help robotic tasks in low-light settings, by equipping robotic hands with tactile sensors. We can leverage our pretrained cross-modal manifold to recognize objects and even operate objects when vision is impaired. With our trained shared latent space we train a classifier to classify the corresponding
> object category and manipulation type from the signal sequence. We withhold some latent codes from the classifier to test the baseline performance on human data. We use test-time optimization to obtain a latent code in the shared embedding space using the robotic tactile data. We show the performance in Table. 2 in Appendix. We notice that human manipulation and robotic manipulation share a significant amount of transferrable and common knowledge.
>
> While these are simple robotic applications of our proposed dataset and method, other more advanced robotic applications are possible. We focus on learning a joint representation for different modalities of signals and leave more challenging robotic tasks as future work
>
> **2. Moreover, it is worth noting that a single video observation could potentially correspond to a wide range of tactile signals, such as variations in the force applied when touching dough. Regrettably, the study does not appear to account for the inherent multimodality in the distribution of data in this respect;**
>
> Thank you for the question. Our method is able to handle and accommodate variational inference / multimodal predictions. This is actually the strength of our method. we have modified the manuscript to reflect this in section 4.3. Specifically, our framework learns the joint manifold of visual and tactile signals. This allows us to handle multimodal settings, as each possible pairing is in the manifold. This can be extracted by running test-time optimization with different latent initializations. By initializing different latent codes for test-time optimization, we can allow variational inference for a fixed sequence that is given. We added an experiment to reflect shown in the added Section A3 in the manuscript.
>
> Even though touching dough is a very interesting and challenging topic indeed, our work primarily focuses on articulated object manipulation and cross-modal inference on these objects. We leave touching dough an interesting future work.
>
>
> **3. The video prediction results for new instances and new categories seem not promising.**
>
> Thank you for the comment. Generalizing manipulation to new categories of objects is a very challenging task. When compared to the baseline methods, our proposed method achieves much better generalizability. With more data, our proposed method should do even better. We hope our work can shed some light on the research direction of generalizing manipulation to novel instances and categories.

---

> > ### Author Response · Authors · 2023-11-20
> >
> > **4. Providing additional technical details would enhance the reproducibility of the work, e.g., model architecture, training details, etc.**
> >
> > Thank you for this feedback, we have updated our manuscript to include more detail on training and implementation, reflected in Section.A5 and A6 in the Appendix section.

---

> > > ### Comment · Reviewer_RFKG · 2023-11-22
> > >
> > > Thanks for the authors’ response, which clarifies the majority of my inquiries.

---

> > > > ### Author Response · Authors · 2023-11-22
> > > >
> > > > Hi reviewer RFKG,
> > > >
> > > > We appreciate the response. We are wondering if there is still anything that keeps your score at boarderline reject? We would love to help to further address your concerns.
> > > >
> > > > Thanks again.

---

### Official Review · Reviewer_h9Bd · 2023-10-23

**Soundness:** 3 good
**Presentation:** 2 fair
**Contribution:** 3 good
**Rating:** 6
**Confidence:** 3

**Summary:**

This paper focuses on understanding everyday objects and tools from a manipulation standpoint. The authors have constructed a multi-modal visual-tactile dataset, consisting of paired full-hand force pressure maps and manipulation videos. Additionally, they introduce a unique method to learn a cross-modal latent manifold. This manifold facilitates cross-modal prediction and uncovers latent structures in various data modalities. The extensive experiments establish the efficacy of the proposed approach approach.

**Strengths:**

1. This paper tackles a common issue, manifold learning, through a pragmatic lens within robotics applications. The study aims to address multi-modal learning problems in the visual-tactile sensory observation context, a highly practical setup for manipulation tasks. The proposed representation learning method can be beneficial for a multitude of downstream applications within visuotactile learning in robotics.

2. The method proposed in the paper is straightforward, suggesting that it does not place a heavy computational load on the system.

3. The authors have gathered a substantial paired dataset for visual and tactile signals. If made publicly accessible, this dataset could prove to be a valuable resource for further research.

**Weaknesses:**

1. The method operates under the assumption that the sum of shape and tactile information equates to visual information. This assumption is manifested in the authors' approach of creating video latents by combining the latents of manipulation and the latents of canonical shapes. However, the tactile sequence may also encapsulate the object's geometric information. As suggested in the referenced paper 'Learning human–environment interactions using conformal tactile textiles,' tactile information can be employed to classify object geometry. Consequently, it's worth questioning the efficacy of combining shape and tactile embeddings to produce the video embedding.

2. The cross-modality query necessitates an optimization process. Therefore, it's crucial to provide information regarding the time cost of these experiments. For instance, how much time would be required to employ the neural field in this inverse manner?

3. The absence of videos in the paper is a notable limitation. Including video content could significantly enhance the understanding of the tasks and experiments conducted in the study.

**Questions:**

1. Do you want to claim the dataset as one of the contribution? In another word, would you open source the dataset once the paper is accepted?

2. Could you clarify the symbol $\gamma$ used in Equation 2? I was unable to locate a definition for it within the text.

3. Your elaboration on $I_i$ and $I_j$ would be appreciated, specifically in relation to the following sentence. How is the distance within the space of $I$ quantified, and how is the subtraction operation defined in Eq3 for $I_i$ and $I_j$, given that $I$ is a three-modality tuple? While I recognize that the manifold isometry loss is a standard loss in the manifold learning field, I would like to confirm if the subtraction operation is a simple reduction operation in the raw data format.
> any two samples sampled from the signal agnostic manifold $\{z_i, z_j \} ⊆ M$ respects the distance between the samples $I_i, I_j$

Minor issues:
1. There appears to be a typographical error in the first line of the 'Dataset: Data Collection Setup' section – the citation parentheses are empty.
2. It would enhance clarity if Figure 1 was referenced in Section 4.2 and if further details about the components in the figure were provided. This issue is also applicable to Figure 2.
3. Please use last names when citing authors. For instance, it should be 'Chen et al.' instead of 'Peter et al.'.

---

> ### Author Response · Authors · 2023-11-20
> **Updated our draft to reflect 1) intuition on our proposed method 2) test-time optimization time cost, 3) added videos to supplementary website, 4) more details our open-source datset, 5) clarification of equations, 6) corrected typos.**
>
> We thank the reviewer for the constructive feedback. We have updated our draft to reflect 1) intuition on the method 2) test-time optimization time, 3) added videos to supplementary website, 4) more details on dataset to be open-sourced, 5) clarification of equations, 6) corrected typos.
>
> **1. The method operates under the assumption that the sum of shape and tactile information equates to visual information. This assumption is manifested in the authors' approach of creating video latents by combining the latents of manipulation and the latents of canonical shapes. However, the tactile sequence may also encapsulate the object's geometric information. As suggested in the referenced paper 'Learning human–environment interactions using conformal tactile textiles,' tactile information can be employed to classify object geometry. Consequently, it's worth questioning the efficacy of combining shape and tactile embeddings to produce the video embedding.**
>
> Thank you for the feedback. Reconstruction and classification tasks are fundamentally different. Classification can rely on local geometric information to predict object classes when objects are sufficiently different, as shown in the paper “Learning…. “, whereas constructing the video sequence requires detailed information on the color of the object, the full geometry of the object, including the areas of the object that is not touched. The tactile force maps obtained from the tactile glove are not informative about the color of the object, the untouched area of the object, and the texture appearance of the object.
> We thank the reviewer for the comment. We have updated the writing of our paper from “tactile sequences contain less information about the object geometry”  **to** “tactile sequences contain limited information about the object appearance. It is not informative about the color of the object, the untouched area of the object, and the texture appearance of the object. We leverage complementary information from canonical images that contains complementary information on the color/texture/full geometry of the objects to construct video sequences.“
>
> Additionally, under our projection framework, shared information in both tactile and shape will be projected in both latent spaces, so this is not a problem.
>
> **2. The cross-modality query necessitates an optimization process. Therefore, it's crucial to provide information regarding the time cost of these experiments. For instance, how much time would be required to employ the neural field in this inverse manner?**
>
> We thank the reviewer for bringing this to our attention. We have included section A6 for more detail on the time for test time optimization.
>
> Specifically, for test-time optimization of the latent code of our proposed approach, we use Adam optimizer to run 1000 time steps with a batch size of 32. For 1000 time steps, we use 56 seconds in total, 17.87 iteration/second, on average every example uses 1.75 seconds. With better computational resources or faster optimizers, the inference speed can be faster.
>
>
> **3. The absence of videos in the paper is a notable limitation. Including video content could significantly enhance the understanding of the tasks and experiments conducted in the study.**
>
> Thank you for this constructive feedback. We have updated our website to include videos on our [supplementary website](https://sites.google.com/view/iclr-submission-force-vision)
>
> **4. Will the dataset be open-sourced once the paper is accepted?**
>
> Thank you for the question, and yes, we will open-source the dataset once the paper is accepted. We have updated our details on the dataset that will be open-sourced for public use.
>
> **5. Could you clarify the symbol $\gamma$ used in Equation 2? I was unable to locate a definition for it within the text.**
>
> Thank you for bringing this up. This is a typo. We have updated our draft with the correct formulation. Specifically, $L_{Rec} = | \phi_{c} (z_{c, i}) - c_i|^2 + | \phi_{h} (z_{h, i}) - h_i|^2 + | \phi_{x} (z_{x, i}) - x_i|^2 $

---

> ### Author Response · Authors · 2023-11-20
>
> **6. Your elaboration on $\mathcal{I}_i$ and $\mathcal{I}_j$ would be appreciated, specifically in relation to the following sentence. How is the distance within the space of $I$ quantified, and how is the subtraction operation defined in Eq3 for $I_j$ and $I_i$, given that $I$ is a three-modality tuple? While I recognize that the manifold isometry loss is a standard loss in the manifold learning field, I would like to confirm if the subtraction operation is a simple reduction operation in the raw data format.**
>
> Thank you for this question. We have updated our draft for more clarity on the manifold isotropy loss. Specifically, we allow the distance between two sampled latent codes to be equal to the distance between the explicit signals expressed by the latent codes, using $d_z(i, j)$ to denote the latent distance between two latent codes $z_i,z_j$, and use $d_\mathcal{I}(i, j),\mathcal{I} \in \{x, c, h\}.$ to denote distance between 2 signals:  $\| c_i - c_j \|$, $\| x_i - x_j \|$, or $\| h_i - h_j \|$. In our formulation:
>
> $$L_{Iso} = d_z(i, j) - d_{\mathcal{I} (i, j)} \\ =  \left(d_z(z_{c,i}, z_{c, j}) - d_{\mathcal{I}} (c_i, c_j)\right) +\left(d_z(z_{h,i}, z_{h, j}) - d_{\mathcal{I}} (h_i, h_j)\right)+ \left(d_z(z_{x,i}, z_{x, j}) - d_{\mathcal{I}} (x_i, x_j)\right)$$
> $$= \left( \| \|(z_{c, i}) - (z_{c, j})\| - \| c_i - c_j \|   \| \right)+  \left(\| \|(z_{x, i}) - (z_{x, j})\| - \| x_i - x_j \|   \| \right) + \left(\| \|(z_{h, i}) - (z_{h, j})\| - \| h_i - h_j \|  \| \right) $$
>  $$= \left( \| \|\Gamma_c({z_i}) - \Gamma_c({z_j})\| - \| c_i - c_j \|   \| \right)+  \left(\| \|\Gamma_x({z_i}) - \Gamma_x({z_j})\| - \| c_i - c_j \|   \| \right) + \left(\| \|\Gamma_h({z_i}) - \Gamma_h({z_j})\| - \| h_i - h_j \|  \| \right) $$
>
> We have updated our manuscript to reflect this in equations 4 and 5 in Section 4.2.
>
> **7. typos**
>
> We thank the reviewer for pointing out our typos. We have corrected these issues and reflected in our updated manuscript.

---

> > ### Comment · Reviewer_h9Bd · 2023-11-22
> > **Replay to Author Rebuttal**
> >
> > I appreciate the authors' substantial effort in addressing the points raised during the review. The detailed explanation provided for the derivation has enhanced my understanding of the issue at hand. Additionally, the new webpage is truly impressive. While I am inclined towards the acceptance of this paper, considering the application and overall novelty, it's challenging to advocate for a strong acceptance.

---

### Official Review · Reviewer_u4UQ · 2023-10-29

**Soundness:** 3 good
**Presentation:** 3 good
**Contribution:** 3 good
**Rating:** 6
**Confidence:** 2

**Summary:**

This paper purposed a force maps and RGB paird visual-tactile dataset. And further purpose to first represent each signal in a shared latent space, and then project the global manifold to local submanifold for each signal for reconstruction. The results demonstrate the effectiveness of purposed method.

**Strengths:**

1. This paper tactles a more challenging visual-tactile prediction task which is harder compare to previous works.
2. The design of preoject global manifold to local submanifold force the model to capture different signals, and to futher incorporate with the test time optimization method to improve the prediction results.
3. The experiments are comprehensive.The TSNE results also show the model learned with some semantic meaningful infomation.

**Weaknesses:**

1. My major concern is since the training set : testing set is aroud 12:1, and only include 4 categoreis, can this method really generalize to unseen objects? How is the diversity of the training and testing set?
2. Presentation with Figure1: It would be better to also draw the process of getting shared latent space in the Figure1 for better understanding, it's quiet hard to undertand how to get a shared latent space from signals that are different dimension, could the author illustrate more about this ? Also projection layer of x seems missing in Figure 1.

**Questions:**

1. If as the author statement, force maps and RGB id many-to-many mapping, what is the advantage of using force maps as tactile signal?  Why different object will not have similar surface texture property? And the challenge of disparity in spatial scale of different signals seems also exist even if it is one-one mapping?
2. How is the robustness of the tactile glove, will it need to calibrate a lot to make sure the tactile data is accurate?
3. How long will it take for test time optimization? Since this might be important for robotic application?

**Details Of Ethics Concerns:**

No ethics concerns.

---

> ### Author Response · Authors · 2023-11-20
> **Updated 1) draft with clarification on dataset, 2) more details on proposed shared latent, 3) 4) details on data calibration 5) robotic experiment**
>
> Thank you for the constructive feedback. We have updated our draft with 1) better clarification of train-test split, 2)clarification on shared manifold construction, 3) discussion and history on force maps as tactile signal, 4) details on calibration, 5) test-time optimization time.
>
> **1. My major concern is since the training set : testing set is aroud 12:1, and only include 4 categoreis, can this method really generalize to unseen objects? How is the diversity of the training and testing set?**
>
> Thank you for raising this concern, we have revised our manuscript for increased clarity on the dataset statistics, shown in Section 3.  we provide more details and more description on the details of our dataset and train-test split. Our training and test set is 6:1, with 124,000 frames of data for training, (10,000 + 6,000 + 5,000) frames of data for testing. The details are as follows
>
> - Our training set contains 81 objects spanning 4 different categories, with paired tactile and video recordings of manipulation, containing 124000 frames of data.
> - Our test set is constructed with 3 different subsets:
>     - in-distribution test-set: constructed with withheld sequences of the same 81 objects seen during training, containing 10,000 frames of data.
>     - in-category test-set: contains withheld 4 objects unseen during training, containing around 6,000 frames of data.
>     - out-of-category test set: we collect data on 4 objects of unseen categories, containing 5,000 frames of data.
>
> We test our proposed method on novel instances of a seen category as well as a novel category as shown in Section 5.3 in our paper. We do want to note that with more data, our proposed method can achieve better generalization.
>
> **2. Presentation with Figure1: It would be better to also draw the process of getting shared latent space in the Figure1 for better understanding, how to get a shared latent space from signals that are different dimension, where is projection layer of x?**
>
> Thank you for the constructive feedback. We have updated the figure and writing for more clarity, reflected in Section 4.1 and 4.2 in blue.
>
> With regard to the shared latent space, Existing methods, such as TouchandGo, ObjectFolder, VisGel, etc., employ encoder-decoder techniques where the latent codes are constructed and obtained from encoding a specific modality of information. However, we use an auto-decoder technique, using different branches of decoders to construct different modalities of information (tactile/vision). In this way, we allow gradients from different modalities of information to simultaneously update and optimize the latent embedding. In our implementation, we follow the DeepSDF implementation and randomly initialize our shared latent embedding to be trainable. During training, we sample a latent code $z_i$ to decode to different through different neural fields (coordinate networks), tactile signal $I_h = \phi_h (\Gamma_h(z_i))$ and video signal $I_x = \phi_x(\Gamma_x(z_i))$.
>
> In order to handle information of different dimensions, we use neural fields or coordinate networks. As described in Section 3.1 in our paper, neural fields are different from traditional convolution networks whose construction is significantly bound by the dimension of signals. Neural fields take in coordinates $u$ and some latent code $z_{k}$ that contains information about the signal $k \in \{ c, x, h\}$  to decode value at that coordinate $u$.  Formally, the value of the $i$-th sample for signal $k$ at coordinate $c$ is obtained:
> $I_{k, i, u} = \phi_k ( z_{k, i}, u )$.
> In our implementation, we use MLP that takes in input of
> $d_z + d_u$
> dimensional to decode signal values
> $R^{d_{I}}$
>  at coordinate
> $u$, $\phi_j$
> where
> $\phi_j: R^{d_z + d_u} \rightarrow R^{d_I}$.
> For more information and interest in neural fields, please refer to [3]
>
> We have updated our text and visual presentation on the construction of the video projection layer in our manuscript to provide more detail and clarity.
> The video projection layer $\Gamma_x$ is constructed by concatenating the image projection layer and tactile projection layer, For the details of each of our projection layers:
>
> $z_{c, i} = \Gamma_c(z_i), \Gamma_c \in \mathbb{R}^{256 \times 256}$
>
> $z_{h, i} = \Gamma_h(z_i), \Gamma_h \in \mathbb{R}^{256 \times 16}$
>
> $z_{x, i} = \{\Gamma_c; \Gamma_h\}(z_i) = \{\Gamma_c(z_i) ; \Gamma_h(z_i)\} = \{z_{c, i} ; z_{h, i}\}$
> or equivalently, $\Gamma_x = \{\Gamma_c; \Gamma_h\} \in \mathbb{R}^{256 \times 272}$
>
> [1] Gao et al. Objectfolder: A dataset of objects with implicit visual, auditory, and tactile representations. CoRL, 2021.
> [2] Yang et al. Touch and go: Learning from human-collected vision and touch. NeuIPS
> [3] Li et al. Connecting touch and vision via cross-modal prediction. CVPR June 2019
> [4] Xie, Y., Neural fields in visual computing and beyond. In *Computer Graphics Forum*.

---

> ### Author Response · Authors · 2023-11-20
>
> **3. If as the author statement, force maps and RGB id many-to-many mapping, what is the advantage of using force maps as tactile signal? Why different object will not have similar surface texture property? And the challenge of disparity in spatial scale of different signals seems also exist even if it is one-one mapping?**
>
> Thank you for the question. Tactile signal might be a slightly overloaded term for referring to different types of signals, such as GelSight surface textures [1, 2, 3] and force responses from touch [5, 6]. To clarify, we use force maps as tactile signals, and we are not concerned with surface textures, by following previous Subramian et al.[5] and Luo et al [6]. Tactile, in general, refers to the responses of our skin when touching and making contact with the external environment. In this work, we only consider force maps instead of surface textures as our touch/haptics signal, because we are interested in different types of object manipulation. For this purpose, force maps are more informative about the manipulation action than knowing the surface texture of various objects. We believe surface texture is a very interesting topic indeed, but it is less relevant to our focus on hand-object manipulation.
>
> The large disparity in spatial scale, in general, incurs challenges. Tasks become more difficult when the relation between different modalities is many-to-many mapping. In the cases with force-map tactile signals, dependent on the deformability of objects, the force map will be very different. In general, there could exist many pairings between RGB images and force maps. Our paper is interested in discovering how we can discover the shared structure across such disparate settings.
>
> [5] Sundaram et al. Learning the signatures of the human grasp using a scalable tactile glove. Nature, 2019
>
> [6] Luo, et al. Learning human–environment interactions using conformal tactile textiles. Nature Electronics, 2021
>
> **4. How is the robustness of the tactile glove, will it need to calibrate a lot to make sure the tactile data is accurate?**
>
> Thank you for the question. we have included more details on the background of the gloves and glove calibration in the dataset section in Appendix section A.1.
> Specifically, the glove only needs to be calibrated once. The glove is made using piezoresistive film. The working principle of this material is when the material stretches, it causes the deformation of carbon nanotubes, which then induces the relative electrical resistance changes and thus difference in voltage readouts. The material in general is sturdy and the material property does not change much with wear and tear. Therefore, the glove only needs to be calibrated once to convert the sensor readouts in voltages to force magnitudes in Newton before use.
>
> **5. How long will it take for test time optimization? Since this might be important for robotic applications?**
>
> We thank the reviewer for bringing this to our attention. We have included section A6 for more detail on the time for test time optimization.
>
> Specifically, for test-time optimization of the latent code of our proposed approach, we use Adam optimizer to run 1000 time steps with a batch size of 32. For 1000 time steps, we use 56 seconds in total, 17.87 iteration/second, on average every example uses 1.75 seconds. With better computational resources or faster optimizers, the inference speed can be faster.

---

> ### Comment · Reviewer_u4UQ · 2023-11-21
>
> Thanks for your reply!
> I have three further questions:
> 1. I am still curious what type of objects you are using for out-of-category tests, can you provide more details on this? (for example, provide the object list of your whole dataset in website)
> 2. Consider your current design, Is this method robust for input video that has a different background? if during training, include a video image that has a different background, will it has a side effect for training, since the static image may not have information that the video image requires?
> 3. although force map can offer more information about manipulation, will using additional texture can make this question easy, since the texture will make object classification easier, which seems can help map force and RGB?
>
> Also for question 5: such time seems hard for real-time robotic manipulation, which might need to be improved.
>
> Although there are still few. question I am curious, these question may not eliminate the contribution of this work, thus, I would like to raise my score.

---

> > ### Author Response · Authors · 2023-11-22
> >
> > Hi Reviewer u4UQ,
> >
> > Thank you for your questions.
> >
> > **What type of objects you are using for out-of-category tests, can you provide more details on this?**
> >
> > For our out-of-category objects, we used clamps. We will add a list of objects to our website.
> >
> > **Is this method robust for input video that has a different background? if during training, include a video image that has a different background, will it has a side effect for training, since the static image may not have information that the video image requires?**
> >
> > Thank you for this question. Applying data augmentation during training can improve robustness of our method as well as other learning-based methods. During training, data augmentation techniques such as color randomization, cropping, and masking can be applied to the input data to increase the data distribution help methods to become more robust to handling video with different background. We note that this technique can be used with all learning-based methods, and the results reported in the paper for all methods uses the same data preprocessing.
> >
> > **although force map can offer more information about manipulation, will using additional texture can make this question easy, since the texture will make object classification easier, which seems can help map force and RGB?**
> >
> > Thank you for this question. The goal of our work is not object classification instead we focus on learning manipulation of articulated bodies. Manipulation relies heavily on object weight, size, motion axis, articulation type, and etc. Texture is certainly an interesting signal modality, but it is not very informative on these aspects, and thus is less helpful when it comes to the kind of manipulation we consider in our work. If object classification is to be leveraged, video and image of the object is likely to be more informative about the object type than the texture, e.g. different types of object can also be made using the same kinds of plastic.
> >
> > **such time seems hard for real-time robotic manipulation, which might need to be improved.**
> >
> > Time cost is a common problem for methods that requires test-time optimization. On robotic manipulation tasks, our method can be helpful when used as a pretraining or learning from demonstration technique. We show in our added experiment in Sec. A2 in Appendix that our method allows human-manipulation data and robotic-manipulation data to be projected in the same space. Our proposed method can then serve as a pretraining technique to warm-start policy-learning for robots and increase sample efficiency for when training RL policies for robotic manipulation tasks, especially consider that robotic manipulation is very data hungry.

---

> ### Comment · Reviewer_u4UQ · 2023-11-22
>
> 1. As you mentioned, articulation type is important for object manipulation, and the same object type usually has the same articulation type, from my perspective, if you know the object type, you will also know more information about how to manipulate it.
> 2. I am also curious about how to use such a method for RL training. For example, for every RL step, you will get an new image, and then you have to do optimization to get correspond latent, how can you avoid these?

---

> ### Author Response · Authors · 2023-11-22
>
> Thank you for the comment.
>
> **As you mentioned, articulation type is important for object manipulation, and the same object type usually has the same articulation type, from my perspective, if you know the object type, you will also know more information about how to manipulate it.**
>
> Indeed, knowing the object type will help with knowing the general articulation type. However, in addition to object type, the actual action that will induce the desired articulation varies based on the size of the object, and physical properties of parts inside the object (e.g. stiffness or strength of the springs), and etc. Force maps are more informative on these aspects as compared to texture. The video sequences used in the method are very informative about the object type, and as shown in the TSNE of the latent visualization, the learned latent manifold clearly clusters data based on both the object type and manipulation type.
>
> **I am also curious about how to use such a method for RL training. For example, for every RL step, you will get a new image, and then you have to do optimization to get the corresponding latent, how can you avoid these?**
>
> The most naive way to use our method in a robotic setting is to use it as a pretraining strategy for RL. For example, each latent code in the pre-trained latent embedding can be projected and decoded into state spaces for the robot arm to conduct similar actions as the decoded human manipulation sequences to mimic human manipulation. With such a pretraining stage, sample efficiency can be improved as training the RL policy is now dealing with a subset of possibly higher-success state spaces instead of all possible state spaces.  We leave efforts on policy training for robotic manipulation a future work, and study the hand-object manipulation synergy in the current effort.
>
> Additionally, we want to note that decoding each latent code into video and tactile sequences is instantaneous, the optimization takes a long time because many optimization steps are taken, and our method is still relevant as long as the test-time optimization is not used during the online testing stage.

---

### Official Review · Reviewer_ksmF · 2023-10-30

**Soundness:** 3 good
**Presentation:** 3 good
**Contribution:** 2 fair
**Rating:** 6
**Confidence:** 3

**Summary:**

The authors have curated a unique visual-tactile dataset and introduced a manifold algorithm to explore the cross-modal relationship between objects and their manipulation. By visualizing the cross-modal latent structures, they showcase that their approach outperforms current methods and effectively generalizes manipulations to unfamiliar objects.

**Strengths:**

1. The paper is articulately written, offering clarity and ease of comprehension, making it accessible even to readers unfamiliar with the subject matter.
2. A significant contribution of this research is the introduction of a novel visual-tactile dataset, especially noteworthy given the limited datasets available in this domain.
3. The innovative manifold learning approach presented has the potential to pave the way for subsequent research.
4. Through experiments, the paper effectively showcases the promise of the cross-modal retrieval, prediction, and the latent structure. Compared to existing methodologies, the proposed approach holds considerable promise.

**Weaknesses:**

1. The dataset would benefit from enhanced visualization and in-depth details, possibly within the appendix.
2. There's a typographical error on page 5 after equation 2; "Additioanlly" should be corrected to "Additionally."
3. Based on observations from figures 3 and 4, the sequences appear to have minimal variation across different frames. Displaying greater variation would add value. Additionally, considering a baseline that utilizes only the initial frame, as opposed to the entire sequence data, could provide intriguing insights.

**Questions:**

Please see the weakness above.

---

> ### Author Response · Authors · 2023-11-20
> **Updated draft and website 1)  include more visuals and videos, 2) corrected typos, and 3) more experimental results**
>
> Thank you for your detailed comments. We have updated our manuscript to include more visuals and videos, corrected typos, and added more results with more frame-wise variations.
>
> **1) The dataset would benefit from enhanced visualization and in-depth details, possibly within the appendix.**
>
> Thank you for this feedback. We have updated our manuscript to include more details about our dataset both in Section A2 in the Appendix and on our [supplementary website](https://sites.google.com/view/iclr-submission-force-vision/). We added a demo figure of video sequences of our tactile and RGB signal from 12 objects sampled from our dataset, as well as some video sequences of experiments. We have also added more details about our dataset collection and dataset statistics in Section 3 of our paper. We also added a Section to cover more background on the tactile glove and sensor calibration. We hope these modifications improve the clarity of our paper about our dataset.
>
> **2) There's a typographical error on page 5 after equation 2; "Additioanlly" should be corrected to "Additionally."**
>
> We have corrected this typo. Thank you for bringing this to our attention.
>
> **3) Based on observations from figures 3 and 4, the sequences appear to have minimal variation across different frames. Displaying greater variation would add value. Additionally, considering a baseline that utilizes only the initial frame, as opposed to the entire sequence data, could provide intriguing insights.**
>
> Thank you for the feedback. We hope that the video demo of our dataset on our supplementary webpage helps provide a better illustration of the signal variation across different frames.
>
> We added a test that only the first frame as the test-time optimization energy signal and added it to our ablation experiments. Empirically, we observe that it only using first frame decreases cross-modal retrieval accuracy. We believe that this difference is caused by the increase of variational inference, or in other words, with less optimization signal, there are now more local minima during test-time optimization that fit the optimization objective. More details can be found in Sec. A4 in our appendix. We would appreciate more details on the kind of insights or the kind of first frame setup you may be looking for.

---

> > ### Comment · Reviewer_ksmF · 2023-11-20
> >
> > Thanks for your reply! My concerns have been addressed, but I will keep the original rating given the whole contribution.

---

### Author Response · Authors · 2023-11-20
**Main Updates for Rebuttal**

We thank all reviewers and ACs for their time and effort in reviewing our paper and for their constructive feedback and

We are glad that the reviewers find the following contributions of our work:

**Dataset**: Dataset is novel and makes up for the lack of such a dataset in the domain (ksmF).  Our paired dataset for visual and tactile signals is substantial and could prove to be a valuable resource for further research (h9Bd, RFKG)

**Method**: Our proposed method is innovative (ksmF, u4UQ). Useful for downstream robotic applications (h9Bd).

**Experiments**: Results are comprehensive and show cross-modal retrieval, prediction and TSNE for latent structure(ksmF,u4UQ).

**Presentation**: The paper is articulately written, offering clarity and ease of comprehension (ksmF). Paper is clear and easy-to-follow (RFKG)

---

We will address reviewers' concerns in the individual responses. We also revised our manuscript according to the reviewers' suggestions, and we would like to note that we made the following **major updates to our manuscript** and [supplementary website](https://sites.google.com/view/iclr-submission-force-vision/) following the suggestions by the reviewers:

- Clarification on data: data collection setup (Sec. 3), dataset Statistics (Sec. 3), data visualization (Sec. A1), background on glove and calibration (Sec. A1) -- (ksmF, u4UQ)
- Clarified method writing: a detailed explanation of shared manifold construction (Sec. 4.1), construction of video manifold projection (Sec 4.2), clarify manifold isotropy loss (4.2) -- (u4UQ, h9Bd)
- Added Experiment:
    - Robotic-inspired downstream applications (Sec. A2) -- (RFKG)
    - Tactile variational prediction (Sec. A3) -- (RFKG)
    - Ablation test on first-frame data (Sec. A4) -- (ksmF)
- Added Technical Detail:
    - Implementation Detail (Sec. A5) -- (RFKG)
    - Training Detail (Sec. A6) -- (RFKG)
    - Test-time optimization time cost (Sec. A5) -- (RFKG, u4UQ, h9Bd)

---

### Meta-Review · Area_Chair_WYZm · 2023-12-03

**Metareview:**

The paper unanimously received a “marginally above acceptance threshold” rating. The reviewers had various questions regarding the dataset split, disparity in the spatial scale of the different signals, robustness of tactile gloves, etc. During the discussion period, some of the reviewers engaged in a conversation with the authors and raised more questions. The AC reviewed the paper, the reviews, the rebuttal, and the discussions. The AC believes the proposed dataset will be useful for future research in this domain, the experiments are comprehensive, and the paper is easy to follow. Hence, acceptance is recommended.

**Justification For Why Not Higher Score:**

While the dataset is valuable, it is limited to only 4 categories. A more diverse dataset and more analysis on unseen categories would make the paper stronger.

**Justification For Why Not Lower Score:**

The main contribution of the paper is a multi-modal visual-tactile dataset, which is quite useful for advancing research in this domain.

---

### Decision · Program_Chairs · 2024-01-16

Accept (poster)